# Memory-Modular Classification:
# Learning to Generalize with Memory Replacement

**Dahyun Kang**                                                          *dahyun.kang@postech.ac.kr*
*POSTECH*

**Ahmet Iscen**                                                                *iscen@google.com*
*Google DeepMind*

**Eunchan Jo**                                                          *eunchan9029@postech.ac.kr*
**Sua Choi**                                                                *suachoi@postech.ac.kr*
**Minsu Cho**                                                                *mscho@postech.ac.kr*
*POSTECH*

**Cordelia Schmid**                                                          *cordelias@google.com*
*Google DeepMind*

**Reviewed on OpenReview:** *https://openreview.net/forum?id=DcIWOidrg8&*

## Abstract

We propose a novel memory-modular learner for image classification that separates knowledge memorization from reasoning. Our model enables effective generalization to new classes by simply replacing the memory contents, without the need for model retraining. Unlike traditional models that encode both world knowledge and task-specific skills into their weights during training, our model stores knowledge in the external memory of web-crawled image and text data. At inference time, the model dynamically selects relevant content from the memory based on the input image, allowing it to adapt to arbitrary classes by simply replacing the memory contents. The key differentiator is that our learner meta-learns to perform classification tasks with noisy web data from unseen classes, resulting in robust performance across various classification scenarios. Experimental results demonstrate the promising performance and versatility of our approach in handling diverse classification tasks, including zero-shot/few-shot classification of unseen classes, fine-grained classification, and class-incremental classification.

## 1 Introduction

Large-scale neural models have achieved remarkable results when fine-tuned and applied to downstream tasks in computer vision (Kolesnikov et al., 2020; Yuan et al., 2021; Alayrac et al., 2022) and natural language processing (Brown et al., 2020; Touvron et al., 2023). These models are trained on massive datasets using immense computational resources, resulting in a vast number of model parameters that encapsulate both world knowledge and task-specific skills. This complexity poses two challenges. First, it is difficult to determine which knowledge in the training data or learned skills contributes to the model output for a specific task. Second, models cannot directly reflect changes in the ever-growing real world, such as updates to data sources relevant to the target task, without undergoing additional training.

To flexibly adapt to the external world knowledge, recent zero-shot image recognition models (Guu et al., 2020; Hu et al., 2023b) enhances image representations with their relevant data retrieved from an external knowledge source. Such learning method is often called retrieval-augmented learning. This approach allows models to leverage external knowledge sources and efficiently allocate model parameters to focus on reasoning tasks. Although these models have shown promising results in knowledge-intensive applications, such as

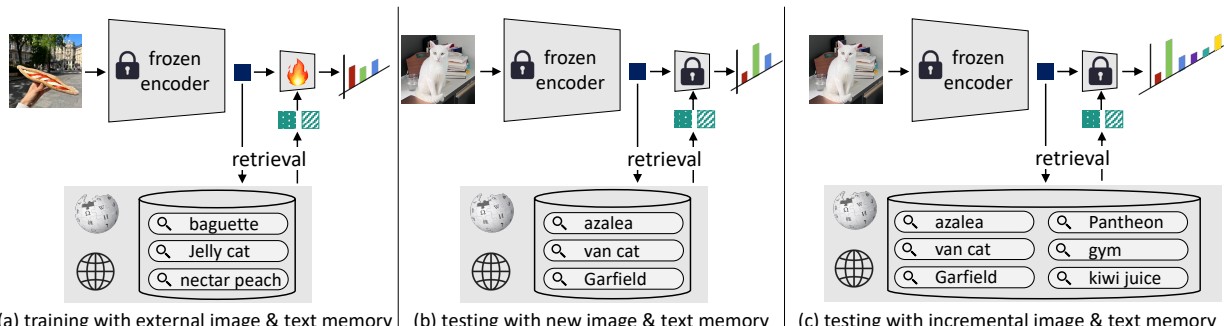

Figure 1: **Training and evaluation stages of MML for web-assisted zero-shot classification.** MML constructs image/text memory with text keyword search on the internet given target classes. The memory provides relevant image/text features which are integrated via a trainable knowledge integration module (a). On evaluation, the memory can be replaced or detached from the model such that MML joins the new knowledge as memory, while the rest of the model remains unchanged. Once trained, MML handles zero-shot classification on unseen classes with memory replacement (b) and incremental classes with memory expansion (c) using the new knowledge collected from web to solve zero-shot classification.

question answering (Gao et al., 2022) and long-tailed classification (Long et al., 2022), their capabilities are limited to a specific target task. Moreover, they assume that the memory content is retained throughout training and testing; the generalizability of the learned models when faced with substantial memory updates or replacements remains unexplored.

In this paper, we introduce a novel learning architecture, the *memory-modular learner* (MML), for image classification. MML leverages an external memory to perform input-adaptive reasoning during the classification process. A key advantage of MML is its ability to generalize with memory replacement, *i.e.*, memory-modular generalization. By simply plugging in new-class content into memory, MML can adapt to novel classification tasks *without requiring any architectural modifications* (Fig. 1). The external memory used by MML is populated by web-crawled images and text obtained by keyword search of the target class names. This approach facilitates the incorporation of up-to-date world knowledge into the memory, ensuring that MML remains applicable as external knowledge evolves. Despite the potential introduction of data noise from web crawling, MML demonstrates robust classification performance in practice. This remarkable robustness allows MML to effectively leverage the noisy memory contents for accurate image classification.

One critical observation of our work is that by representing classification as *metric learning* (Vinyals et al., 2016; Snell et al., 2017), MML becomes less susceptible to overfitting on the specific content of the memory. This allows it to learn more effectively how to perform classification reasoning with arbitrary memory contents. Specifically, we represent the classifier weight vectors, or the *class prototypes*, as the average of representative memory items rather than as learnable parameters. When a new set of classes is given, the class prototypes are immediately computed with the average of the memory items of the highest cross-modal similarity. The input query is classified by the class of the closest class prototype. This design choice allows us to update the memory and adapt to new classes without retraining the entire model. Due to its inherent flexibility, our meta-learned model can handle zero- to multi-shot samples, as well as a variable number of classes with the knowledge collected from web[1]. Experimental results in various scenarios, including zero-shot/few-shot classification of previously unseen classes, fine-grained classification, and class-incremental classification, demonstrate the promising performance of MML.

Our contributions can be summarized as follows.

- We introduce a memory-modular learner (MML) for image classification, that performs adaptive reasoning using external and replaceable memory.

---

[1]We clarify our zero-shot classification approach that accesses to unlabeled web data as *web-assisted zero-shot classification.*

- We investigate the generalizability in adapting to new classes by replacing the memory with related content, without tuning the model weights.

- We provide in-depth analyses on the memory-modular generalization to unseen classes in realistic setups, *i.e.*, using a noisy web-crawled memory.

- We show that MML achieves promising gains in various scenarios such as zero-shot, few-shot, fine-grained, and class-incremental classification by leveraging target-class knowledge collected from web.

## 2 Related work

### 2.1 Few-shot and zero-shot classification with the assistance of external web data

**Few-shot image classification** (Fei-Fei et al., 2006) aims to generalize to arbitrary unseen classes given a few support images from a target class set. The conventional experimental setup of few-shot classification (Vinyals et al., 2016; Allen et al., 2019; Triantafillou et al., 2020; Doersch et al., 2020; Zhang et al., 2020; Kang et al., 2021) assumes at least a few hundred labeled images used for (meta-)training before the actual few-shot inference stage. We, however, adopt a more label-efficient and realistic approach for this task; we train a model with even fewer labeled training samples *e.g.*, $\leq 16$; instead we assume retrieval access to external unannotated data. **Zero-shot classification** (Larochelle et al., 2008; Yu & Aloimonos, 2010) aims for generalization beyond seen classes without the use of few-shot support images for the target classes. Instead, classification is conducted based on non-visual clues such as textual information of the images (Fu et al., 2015; Akata et al., 2016), yes-or-no attributes (Lampert et al., 2013) or the class name in text (Socher et al., 2013) of arbitrary classes. The conventional zero-shot tasks have assumed no use of images from *target classes* during training, but with the advent of web-driven pretrained models, recent "zero-shot" methods (Iscen et al., 2024; Liu et al., 2023) started to use the expression in a more relaxed way, meaning no use of *manually-annotated images* from target classes, thus allowing access to noisy web data. For example, a vision-and-language foundation model named CLIP (Radford et al., 2021) trains image and text encoders with 400 million image-and-caption pairs from internet which likely overlap with standard zero-shot classification benchmark categories. We follow this usage in our paper and leverage web data to leverage the external world knowledge for zero-shot classification. We thus clarify that our approach as **web-assisted zero-shot classification** with the terminology of "shot" denoting the number of *class-annotated images* for each target class.

### 2.2 Image recognition with memory retrieval

One of the earliest works of using an external memory in machine learning is the $k$-nearest neighbor ($k$NN) classifier (Hart, 1968), which retrieves $k$-nearest neighbors from memory for class prediction. Recent work constructs memory from large-scale pre-trained models and performs $k$NN retrieval for class prediction (Khandelwal et al., 2020; Nakata et al., 2022). This straightforward method revisits the potential of external memory for class reasoning, being decoupled from encoder learning (Graves et al., 2014). Image recognition models have also been trained using external image-text paired memory (Jia et al., 2021b; Long et al., 2022; Iscen et al., 2023). Our approach assumes a more weakly-supervised type of memory, collecting image and text memory contents separately. Other external memory-based image recognition work focuses on training multi-modal feature encoders (Wei et al., 2023; Hu et al., 2023b) or training CLIP models with external image-text paired data (Iscen et al., 2024; Liu et al., 2023). One common theme among the existing memory-based models is that they are either trained for a specific task (Long et al., 2022; Hu et al., 2023b; Iscen et al., 2023) or static memory (Iscen et al., 2024). Among them, REVEAL (Hu et al., 2023b) is perhaps most similar to ours. The memory-augmented learning architecture of REVEAL and MML is indeed similar in terms of architecture but different in terms of the role of memory. The memory of REVEAL serves as a general knowledge bank to assist VQA and captioning tasks. On the other hand, the memory of MML contains specifically related contents of the target classes for classification, crawled from web. Therefore, memory contents can be completely replaceable when the target classes are updated – the memory is *modular*. Note that any other previous models do not replace memory contents completely. Also, the difference of general and specific memory also leads to the size difference. REVEAL contains 20.3M memory items

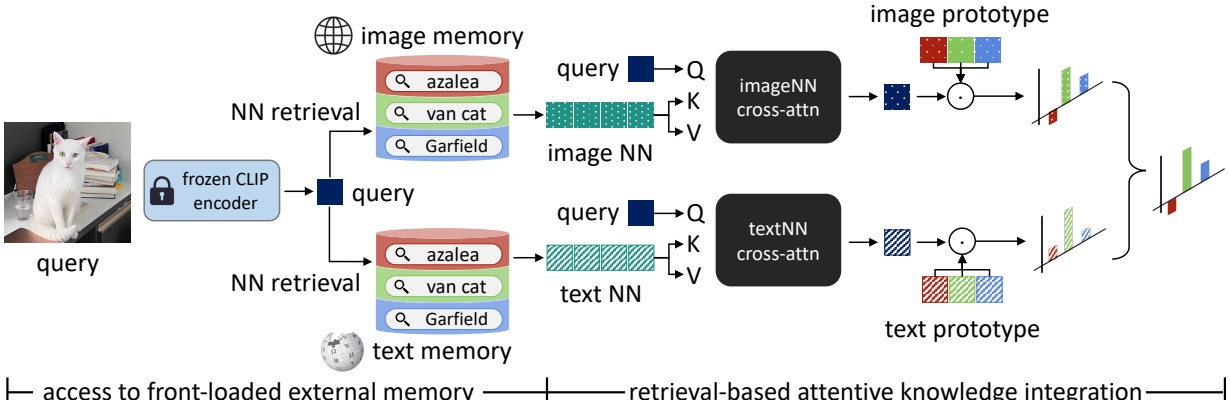

Figure 2: **Memory-modular learner (MML)** constructs image/text memory by web-crawling with text keyword search. Given a query image, its $k$NN features are retrieved from each memory and used for attentive knowledge integration. The class prototypes are constructed with the average of the memory elements of the highest cross-modal similarity. MML derives class reasoning with the nearest neighbors (NNs) from the external memory. This modular memory enables MML to perform web-assisted zero-/few-shot classification on unseen classes by memory replacement and class-incremental classification by memory expansion.

of general image-text pairs. MML requires only 0.7M image and 0.2M text memory items for 1K classes of ImageNet1K, (see Sec. 4.1) which is the 4.4 % size of REVEAL. In contrast to the previous related work, MML aims to generalize beyond a seen class set and modular memory that can be updated at any time. To the best of our knowledge, MML is the first to investigate the memory replacement with new memory contents to tackle unseen-class generalization.

### 2.3 Class-incremental classification

Class-incremental classification (Rebuffi et al., 2017; Zhu et al., 2023) assumes that a set of unseen classes arrives at each stage and aims to classify the input into all known classes given limited access to the old class data. The most critical challenge of this task is catastrophic forgetting, *i.e.*, directly training neural networks with the new-class data leads to significant performance drops in old classes. To address this challenge, recent work (Yan et al., 2021; Wang et al., 2022a; Zhou et al., 2023b; Douillard et al., 2022; Wang et al., 2022b) introduces a memory to store data from previously seen classes as a training source to compile knowledge into a model. In contrast, the memory in MML plays the role of a replaceable and extensible world-knowledge reference. The purpose of memory in these two models are different: the memory of class-incremental learners helps not to forget the previously seen classes (Belouadah & Popescu, 2019; Iscen et al., 2020), however, the memory of MML assists the current classes of interest, which might have not been seen.

## 3 Memory-modular learner

We address the problem of classifying an input image into target classes that are represented by a class name in text, *i.e.*, zero-shot classification, or additional few support images, *i.e.*, few-shot classification. To this end, we introduce a *memory-modular learner* that performs adaptive reasoning using an external memory that is updatable and replaceable. Our memory-modular learner takes advantage of both vision and language modalities using the CLIP encoder (Radford et al., 2021) as a base feature extractor for image and text. Since our method is not restricted to CLIP, any other image-text model, *e.g.* ALIGN (Jia et al., 2021a) or LiT (Zhai et al., 2022), can also be adopted. Figure 2 illustrates the overall architecture of our approach.

The memory-modular learner starts by loading the knowledge memory and generating class prototypes for target classes (Sec. 3.1). These front-loaded memory items and prototypes are all stored as frozen features from a pre-trained image-text encoder. They are replaceable whenever the target classes change or the external knowledge sources are updated. Given an input image, the memory-modular learner accesses

the knowledge memory, retrieves $k$-nearest-neighbor ($k$NN) items, and predicts the corresponding class via cosine-similarity with class prototypes (Sec. 3.2). Since class prototypes are generated immediately from the memory items, the prototype-based classifier can adapt to new target classes of updated memory contents without additional training.

## 3.1 Memory construction and prototype generation

Given target class names or descriptions, we construct the knowledge memory based on available image and text data and generate class prototypes using the memory. As the world knowledge is updated, these memory items can be added or deleted, and even completely replaced, without updating the model weights.

### Knowledge memory

The image memory is constructed using images obtained from keyword searches on the internet. For each target class $c$, images are collected using the class name as the search keyword on a search engine, *e.g.*, Google or Flickr (Kim et al., 2023; Hou et al., 2018). We follow a similar strategy for text memory. In this work, textual information relevant to each target class name is retrieved by querying Wikipedia (Tian et al., 2022; Hu et al., 2023a; Naeem et al., 2023). These web-crawled images and texts may be noisy, but consist of scalable memory contents that reflect the world knowledge. After collecting the relevant images and texts for each target class $c$, we extract their $d$-dimensional features with the image-text encoder, and then store them in the image and text memory: $\mathcal{M}_c^{\text{img}} = \{\boldsymbol{v}_i\}_{i=1}^{N_c^{\text{img}}}$ and $\mathcal{M}_c^{\text{txt}} = \{\boldsymbol{t}_j\}_{j=1}^{N_c^{\text{txt}}}$, respectively.

### Class prototypes

For zero-shot classification, we construct class prototypes based on cross-modal consensus between image and text memory items. For each target class $c$, we first compute the cross-modal cosine similarity $\cos(\cdot, \cdot)$ from each image to all text items of the same class and then select the top-$M$ images with the highest similarity to the texts, *i.e.*, images with high cross-modal consensus. The image prototype for class $c$ is then set to be the average of the $M$ features:

$$\boldsymbol{p}_c^{\text{img}} = \frac{1}{|\mathcal{T}|} \sum_{\boldsymbol{v} \in \mathcal{T}} \boldsymbol{v}, \ \mathcal{T} = \operatorname{argmax}_{\boldsymbol{v}' \in \mathcal{M}_c^{\text{img}}}^{M} \left( \sum_{\boldsymbol{t} \in \mathcal{M}_c^{\text{txt}}} \cos(\boldsymbol{v}', \boldsymbol{t}) \right), \tag{1}$$

where $\operatorname{argmax}_{\boldsymbol{s} \in \mathcal{S}}^{M}(\cdot)$ denotes the top-$M$ operator that returns the best $M$ items from the set $\mathcal{S}$ maximizing the operand function. Based on image-text consensus, this process constructs robust and representative class prototypes from noisy data in the absence of human annotation. Likewise, the text prototype is obtained using the average text-to-image similarity. This zero-shot prototype construction resembles Prototypical Networks (Snell et al., 2017), which averages the $M$ image examples for each class. On the other hand, we build multi-modal prototypes by averaging the representative $M$ samples collected without given annotations. For few-shot classification, *i.e.*, when a few support image samples are available for the target class name, we simply construct class prototypes by averaging the given samples as done in Snell et al. (2017).

### Memory update for adapting to unseen classes

The knowledge memory contents and class prototypes are modular and replaceable. When target classes are updated, *e.g.*, classification of unseen classes or incremental classes, new memory contents are collected to pertain to the new classes. Subsequently, the prototypes for the classes are updated accordingly using Eq. 1.

## 3.2 Reasoning with memory access

Given an input image for classification, we incorporate memory knowledge into reasoning. Items relevant to the input are retrieved from image/text memory and integrated with the input feature through cross-attention. The input is then correlated with the image and text class prototypes. Finally, the predictions from the image and text branches are merged at the logit level for class prediction.

**Memory retrieval**

For an input image feature $\boldsymbol{f}$ extracted from the image encoder, its $k$-nearest-neighbor image items are retrieved based on cosine similarity with all image memory items of all target classes:

$$\mathcal{N}^{\mathrm{img}} = \mathrm{argmax}_{\boldsymbol{v} \in \mathcal{M}^{\mathrm{img}}}^K \left( \frac{\boldsymbol{f} \cdot \boldsymbol{v}}{||\boldsymbol{f}|| \, ||\boldsymbol{v}||} \right), \tag{2}$$

where $\mathcal{M}^{\mathrm{img}} = \cup_c \mathcal{M}_c^{\mathrm{img}}$. The text $k$NNs are also retrieved by querying the image feature to the text memory.

**Attentive knowledge integration**

The knowledge of the retrieved memory items $\mathcal{N}^{\mathrm{img}} = [\boldsymbol{v}_k]_{k=1}^K$ is aggregated by cross-attention (Vaswani et al., 2017; Jaegle et al., 2021) and then integrated with the input embedding $\boldsymbol{f}$. The cross-attention learns to integrate the nearest neighbor (NN) features into the input feature:

$$\boldsymbol{f}^{\mathrm{img}} = \boldsymbol{f} + \sigma \left( \frac{\mathbf{Q}(\boldsymbol{f}) \cdot [\mathbf{K}(\boldsymbol{v}_k)]_{k=1}^K}{\sqrt{d}} \right) [\mathbf{V}(\boldsymbol{v}_k)]_{k=1}^K, \tag{3}$$

where $\mathbf{Q}, \mathbf{K}, \mathbf{V}$ are projection layers with non-linearity, $\sigma$ softmax over $k$ items, and $[\cdot]$ concatenation. Similarly, the same step with the text NN features is performed in parallel. This process can be viewed as a learnable soft NN integration in contrast to the hard majority voting with NNs (Nakata et al., 2022).

**Classification inference**

The resulting embedding is matched against the multi-modal prototypes for all $C$ target classes with cosine similarity $\cos(\cdot, \cdot)$ to produce classification score. The $c$-th class logit $\boldsymbol{z}_c$ is obtained with:

$$\boldsymbol{z}_c = \cos(\boldsymbol{p}_c^{\mathrm{txt}}, \boldsymbol{f}^{\mathrm{txt}}) + \cos(\boldsymbol{p}_c^{\mathrm{img}}, \boldsymbol{f}^{\mathrm{img}}). \tag{4}$$

Final class prediction is conducted simply by taking the class with the highest score.

### 3.3 Training

Our model is trained with cross-entropy loss with one-hot ground-truth class label $\boldsymbol{y}$ and the logit $\boldsymbol{z}$:

$$\mathcal{L} = -\sum_{c=1}^C \boldsymbol{y}_c \log \frac{\exp(\boldsymbol{z}_c/\tau)}{\sum_{c'}^C \exp(\boldsymbol{z}_{c'}/\tau)}, \tag{5}$$

where $\tau$ is a temperature for scaling. Note that we freeze the pre-trained image-text encoder and train the remaining parameters only, *i.e.*, those of attention layers on the image and text branches. The number of training parameters and the frozen CLIP-B/32 is 6.3M and 151M, respectively. Using the frozen pre-trained encoder has three advantages. 1) The pre-trained features provide more reliable similarity for $k$NN retrieval and prototype construction than scratch features, encouraging stable training. 2) Retaining the general pre-trained knowledge, the knowledge integration part converges efficiently with a small amount of data. 3) Most importantly, if the encoder is trained or fine-tuned, then all memory features should be synchronized regularly, while the frozen pre-trained encoder allows us to avoid such extensive computation.

## 4 Experiments

### 4.1 Experimental setup

**Training details**

For the image/text feature extractor, we use the pre-trained CLIP (Radford et al., 2021) and ALIGN (Jia et al., 2021a). Unless specified, CLIP-B/32 is used. For training, we use a batch size of 256, a learning rate

Table 1: Zero-shot cross-dataset transfer. MML is trained with 1 or 4 samples from ImageNet1K coarse-grained classes and tested on 10 fine-grained datasets with zero shot, which is roughly a domain shift scenario.

| method | ImgNet1K | Caltech101 | OxfordPets | Cars | Flowers | Food | Aircraft | SUN | DTD | EuroSAT | UCF | avg. |
|---|---|---|---|---|---|---|---|---|---|---|---|---|
| | | objects | pets | cars | flowers | food | airplanes | scenes | textures | land | actions | |
| zero-shot CLIP (Radford et al., 2021) | 66.7 | 75.9 | 63.6 | 62.9 | 54.7 | 74.5 | 18.2 | 55.3 | 33.3 | 43.0 | 58.7 | 55.2 |
| *k*NN classifier (Nakata et al., 2022) | 55.7 | 87.6 | 72.7 | 68.6 | 75.2 | 75.6 | **29.6** | 56.2 | 33.2 | 37.3 | 63.2 | 59.5 |
| MML (ImageNet1K-1) | 48.3 | 92.6 | 86.4 | 68.1 | 76.2 | 81.8 | 26.2 | 60.0 | 41.6 | 45.6 | 64.2 | 62.8 |
| MML (ImageNet1K-4) | **69.0** | **93.5** | **86.7** | **68.9** | **77.5** | **84.2** | 26.3 | **64.7** | **42.8** | **48.2** | **66.5** | **66.2** |

Table 2: Comparison on zero-shot classification on CUB (Wah et al., 2011) with different backbones. Note that RECO (Iscen et al., 2024) is trained with CC12M.

| method | backbone | accuracy (%) |
|---|---|---|
| CLIP (Radford et al., 2021) | CLIP-B/32 | 70.3 |
| RECO* (Iscen et al., 2024) | CLIP-B/32 | 75.2 |
| MML | CLIP-B/32 | **76.7** |

| method | backbone | accuracy (%) |
|---|---|---|
| CLIP (Radford et al., 2021) | ResNet-101 | 68.8 |
| Yu et al. (2020) | ResNet-101 | 72.4 |
| Xu et al. (2020) | ResNet-101 | 73.8 |
| Chen et al. (2022) | ResNet-101 | 76.1 |
| MML | ResNet-101 | **78.8** |

of $1e^{-6}$ and weight decay of $5e^{-4}$ on a single 2080 Ti or an RTX 3090 GPU for all training and testing. We retrieve 32 NNs from both the image and text memory. We use $M = 16$ for prototype construction and set the logit temperature $\tau = 16$, which is chosen via hyperparameter search. We use three random seeds for drawing few-shot samples randomly and report the average.

**Memory and data**

To construct the external image memory for ImageNet derivatives, we employ a readily available web-crawled image dataset, WebVision ver. 2 (Li et al., 2017). WebVision is collected from Google and Flickr by the keyword search of the 1000 class names of ImageNet1K (Russakovsky et al., 2015). We use the image subset crawled from Google unless otherwise specified. To construct image memory for the other 10 datasets used in Table 1, as no public web-crawled datasets for the corresponding classes are available, we crawl a maximum of 100 images per class from Google with an auto crawler. For text memory, we query Wikipedia for each class name and retrieve the corresponding article text by web crawling. In such a way, the modest length of memory is obtained, *e.g.*, 0.7M images and 0.2M texts for the 1K classes of ImageNet1K, of which *k*NN search is feasible with the PyTorch (Paszke et al., 2017) built-in `topK` module. The dataset details used for zero-/few-shot, fine-grained, and class incremental classification are specified in the corresponding paragraph.

## 4.2 Web-assisted zero-shot classification

First of all, we evaluate our method on zero-shot classification setup, where no labeled images are provided for the target classes. The only information given for the task is a phrased class label for each class, *e.g.*, "van cat", which is used as the search keyword to collect the web-crawled memory.

Datasets: MML is evaluated on single-dataset and cross-dataset zero-shot classification benchmarks. For single-dataset zero-shot classification, ImageNet-S and CUB are used, where the classes of each dataset are split into disjoint sets for few-shot training and zero-shot testing. We adopt the existing zero-shot classification CUB benchmark (Wah et al., 2011; Akata et al., 2013) of which classes are split into 150/50 bird species classes for train/validation. Similarly, we introduce an ImageNet (Russakovsky et al., 2015) split such that it comprises 600/200/200 classes for train/validation/test and call it ImageNet-S (S stands for class *split*). We use 16 images per class for training, *i.e.*, 9.6K training images. For testing on target classes, either zero or a few shots are used for zero- or few-shot classification scenarios. For the cross-dataset setting, we adopt a cross-dataset zero-shot transfer scenario (Zhou et al., 2022), where a model is trained with a few samples from ImageNet1K, *e.g.*, 1 or 4 training samples per 1000 classes, and transferred to 10 fine-grained datasets: Caltech101, OxfordPets, StanfordCars, Flowers102, Food101, FgvcAircraft, SUN397, DTD, EuroSAT, and UCF101. The total classes of these datasets amount to 1,310 classes and their details including the references are found in Table 10.

Table 3: Ablation study on MML components

| model | $k$NN retrieval | learnable integration | ImageNet-S | CUB |
|-------|-----------------|-----------------------|------------|------|
| (a)   |                 |                       | 80.1       | 61.9 |
| (b)   | ✓               |                       | 76.4       | 67.9 |
| (c)   |                 | ✓                     | 75.6       | 60.0 |
| MML   | ✓               | ✓                     | **83.0**   | **75.6** |

Table 4: Effect of different class prototypes

| prototype | ImageNet-S | CUB |
|-----------|------------|------|
| avg of all memory items | 81.2 | 74.4 |
| avg of random memory items | 81.9 | 70.4 |
| text only | 82.7 | 69.1 |
| image only | 82.3 | **76.8** |
| MML (image & text) | **83.0** | 75.6 |

Baselines: The $k$NN classifier (Nakata et al., 2022) retrieves $k$NN of the input from memory[2] and immediately predict the class by majority voting. Zero-shot CLIP/ALIGN extracts text embeddings of the text class names in the predefined templates, *e.g.*, a photo of a van cat, and matches them against the input image embedding. Three state-of-the-art zero-shot models (Yu et al., 2020; Xu et al., 2020; Chen et al., 2022) are also compared, which are trained with the total 8885 annotated images and text attributes of CUB.

Results: Table 1 compares zero-shot baselines and MML on cross-dataset transfer. MML is trained with a few ImageNet samples but exhibits great performance on other datasets with extreme domain shifts, *e.g.*, from classifying general objects (Russakovsky et al., 2015) to land (Helber et al., 2019), *by simply replacing the memory* with the web-crawled domain-related knowledge. In particular, compared with the $k$NN classifier, which is uni-modal and non-learnable, our method meta-learns to integrate the multi-modal $k$NNs and effectively transfers to unseen classes. Table 2 compares MML and other zero-shot models on the zero-shot CUB benchmark (Akata et al., 2013) with 150 training classes and 50 test classes, where MML demonstrates its outstanding effectiveness compared to other models. While the existing models train with full training images and ground-truth attribute annotations (*e.g.*, eye colors), MML learns with *zero* human-annotated attributes, but shows great performance based on integrating the retrieved knowledge from the external memory. Comparing CLIP and ours examines the *significant advantage of external memory access and knowledge integration* where ours obtains a 7.6-11.0 % point accuracy improvement. Plus, we examine the efficacy of different backbones in Table 15 in Appendix, where MML consistently outperforms the others. The following paragraphs continue with more analyses and ablation studies on zero-shot classification.

### 4.3   Model analyses

We present the analyses of the model components. All experiments are based on CLIP ViT-B unless specified.

**Ablation study on model components**

Table 3 presents the ablation study of the main model components of MML. The first model (a) is a zero-shot prototype classifier. When the $k$NN retrieval is added without the learnable $k$NN integration, the model (b) corresponds to the $k$NN classifier (Nakata et al., 2022), which is beneficial on CUB compared to the model (a). The model (c) examines the learnable integration of the cross-attention module without $k$NN retrieval, thus transforming the input feature with the learnable self-attention. The worst result of (c) implies that the additional cross-attention is even harmful without the proper source of $k$NN knowledge integration. The last row with the two components (MML) achieves the highest performance on the two datasets.

**Effect of different class prototypes**

Table 4 compares different methods to build class prototypes. We first try to naïvely average all the contents in each memory to obtain class prototypes without using the top-$M$ operator (Eq. 1). This average aggregation is likely to include plenty of unfiltered data noise, resulting in poor performance on both datasets. Next, "avg of random memory items" randomly selects the same number of items with the proposed cross-modal prototype method and averages them per class. This noisy and unrepresentative prototype leads poor classification. We also attempt to use the single-modality class prototype. The image prototype is more helpful than the text prototype on CUB and the reverse on ImageNet-S, suggesting that the efficacy of the image and text prototype can be dependent on target dataset characteristics. All these methods do not use image-text

---

[2]The original work (Nakata et al., 2022) leverages annotated datasets such as ImageNet1K as image memory, which is expensive to be used as memory. We thus replace it with the noisy web-crawled memory for reproduction.

Table 5: Effect of different memory types

| retrieval from | ImageNet-S | CUB |
|---|---|---|
| no $k$NN | 75.6 | 60.0 |
| text memory | 82.3 | 69.4 |
| image memory | 76.2 | 71.2 |
| unified memory | 76.8 | 71.3 |
| MML (separate memory) | **83.0** | **75.6** |

Figure 3: Effect of memory size on ImageNet-S

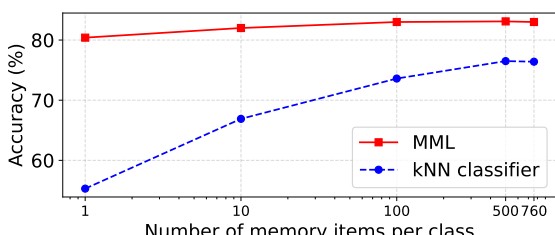

Table 6: Memory content robustness of MML. Memory replacement at testing time from one memory content to another.

| image mem at train
image mem at test | WebVision (WV) Google
WV Google → WV Flickr | WebVision (WV) Flickr
WV Flickr → WV Google |
|---|---|---|
| ImageNet-S | 83.0 → 83.1 (+0.1) | 83.2 → 83.2 (−0.0) |

| text mem at train
text mem at test | Wikipedia
Wikipedia → text thumbnails | text thumbnails
text thumbnails → Wikipedia |
|---|---|---|
| ImageNet-S | 83.0 → 82.7 (−0.3) | 83.2 → 83.0 (−0.2) |

Table 7: Memory content robustness comparison with different noise levels on classification (class) and retrieval

| memory
noiseness | $k$NN retrieval
rec@1 | rec@16 | class
acc. |
|---|---|---|---|
| no memory | - | - | 75.6 |
| noisy (WebVision) | 65.5 | 90.3 | 83.0 |
| clean (ImageNet1K) | 66.4 | 93.5 | 84.7 |

consensus while our method carefully selects memory items that exhibits the high cross-modal similarity for constructing class prototype. In this way, the prototype is comprised of the representative class data and also avoids potential data noise. Using the multi-modal prototypes, our model achieves robust performance.

**Effect of different memory types**

Table 5 validates our dual-branch image and text memory. The "no $k$NN" baseline has the same architecture as the proposed model, but instead, it feeds the input feature for the key and value inputs in replace of the $k$NNs, *i.e.*, the query feature is shared with Q, K, V in Figure 2. This baseline exhibits the lowest performance and signifies the importance of the $k$NN knowledge integration. Next, we ablate either image or text memory. It is noticed that the model using only the image memory is more effective than the one using the text memory on CUB, while this trend is reversed on ImageNet-S. The opposite trend suggests that the vast and detailed visual knowledge collected from the internet is beneficial for fine-grained image classification, on the other hand, textual information is useful for coarse-grained classification of general objects. Lastly, we merge the image and text memory contents and then retrieve the modality-agnostic $k$NN features, which are then passed to a single knowledge integration branch. We observe that the majority of $k$NNs are from the image memory, thus closely matching the performance of the image-memory model. To effectively interact with multi-modal $k$NNs, we choose to separate the image/text memories. This result signifies that the dual-branch multi-modal knowledge integration is crucial in zero-shot unseen class generalization.

**Effect of memory size**

To validate the size of the memory, we set the memory size per class from 1 to full for two retrieval-based models, the $k$NN majority voting classifier (Nakata et al., 2022) and MML, and verify the performance growth in Figure 3 and Table 12. While the larger memory is more helpful, the performance reaches plateau with the abundant memory size. We thus claim that MML requires a moderated size of the retrieval pool as the retrieval and knowledge integration effectively incorporate the useful data from the noisy data source.

**Robustness of memory**

Table 6 shows that MML does not overfit to certain memory contents and performs robustly to different memory contents with little loss of performance. Note that MML already makes unseen class predictions with completely new memory contents of the new classes at the zero-shot testing phase (Tables 1-2). This experiment further examines whether the model performs robustly when *different instances of the same classes* are plugged into the memory. We equip a pair of image/text data collection from two different

| Query image | Retrieved images | Retrieved texts | | |
|---|---|---|---|---|

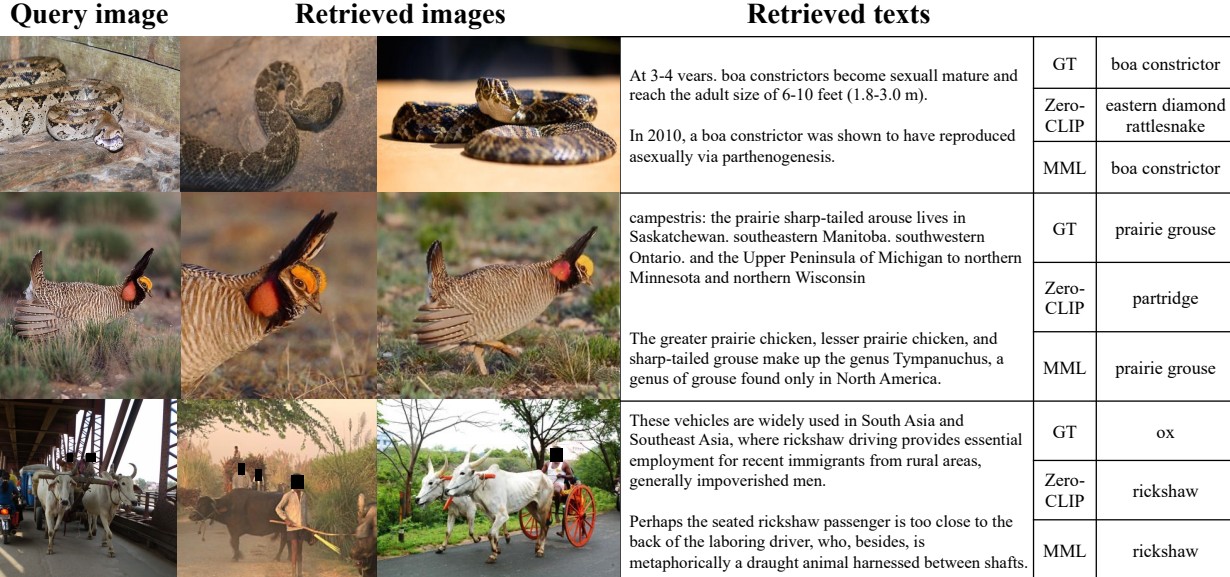

Figure 4: Examples of a query, image 2NNs and text 2NNs. Human faces are anonymized for visualization.

sources of the same target classes. For image memory, we use the two disjoint sets from the WebVision (WV) dataset; one set is obtained through Google crawling and the other from Flickr, where both of them are from ImageNet-S classes. For text memory, we use the text articles from Wikipedia and the text thumbnails of Google text keyword search. Once MML is trained with one source, we test it by plugging the two different memory sources. The results show that the model exhibits a marginal performance gap when replacing the test memory from one to another and also proves the modular property of the memory.

Table 7 presents the robustness of the memory when compared with the clean (human-annotated) and the noisy (web-crawled) memory. The use of web data inherently introduces the trade-off between avoiding additional human annotation and data noise. Retrieval-based learning of MML is also for noise reduction. The retrieval-based knowledge integration remedies such problem by selecting the nearest, *i.e.*, the most relevant, samples to integrate them to establish the data and classifier prototype representation. To quantify the noisiness of the memory data pool, we also present the recall of retrieved items with the standard retrieval metric of recall@$K$ (rec@$K$) (Jégou et al., 2011). The recall@$K$ returns 1 if any instances from the ground-truth class are included in the $k$NN and 0 otherwise. Although the clean image memory enhances mid-level retrieval and end-task classification accuracy, the noisy memory model achieves comparable results to the upper bound. This experiment supports our modeling choice — utilizing web-crawled images as external memory — is reasonably effective and label-efficient compared to fully annotated memory.

**Visualization of multi-modal $k$NN**

Figure 4 visualizes the retrieved two image nearest neighbors (NNs) and two text NNs of the given input as well as the zero-shot CLIP prediction. Note that the images and texts in the example are independently retrieved from each memory. We notice that the image NNs often contain the query's noticeable visual patterns. From the text NNs, we observe that retrieved texts often contain synonymous keywords, *e.g.*, the scientific names of animals. The last example with cows contains multiple objects hence ambiguously class-labeled. In this case, MML is able to retrieve semantically related images and predicts a reasonable class than the ground truth.

Table 8: Few-shot classification on ImageNet-S

| method | 4-shot | 16-shot |
|---|---|---|
| linear-prob CLIP (Radford et al., 2021) | 72.1 | 80.6 |
| ProtoNet (Snell et al., 2017) | 76.4 | 76.5 |
| RAC (Long et al., 2022) | 66.8 | 78.1 |
| $k$NN classifier (Nakata et al., 2022) | 77.2 | 77.2 |
| MML | **82.8** | **83.5** |

Table 9: Few-shot to many-shot classification. TTT stands for test-time training with the given data (shots).

| methods | TTT | 4 | 16 | 64 | 128 | 256 |
|---|---|---|---|---|---|---|
| (a) linear prob | ✓ | 72.1 | 80.6 | 85.5 | 86.9 | 87.3 |
| (b) MML | | 82.8 | 83.5 | 85.7 | 85.7 | 85.8 |
| (c) MML* | ✓ | **83.4** | **87.0** | **87.4** | **87.9** | **88.3** |
| (b) - (a) gap | | +10.8 | +2.9 | +0.2 | -1.2 | -1.5 |
| (c) - (a) gap | | +11.4 | +6.4 | +1.9 | +1.0 | +1.0 |

## 4.4 Application to other classification setups

**Few-shot to many-shot image classification**

Problem setup: Few-shot classification (Fei-Fei et al., 2006; Vinyals et al., 2016) represents unseen classes with few-shot image samples for each target class during testing. We reuse ImageNet-S to make it a few-shot classification scenario by allowing access to additional 4 or 16 labeled images during validation and testing.

Baselines: Linear prob is the simplest few-shot classification baseline (Chen et al., 2019), where we add a class-length linear layer on top of the frozen backbone and train it with the given target class few-shot examples. ProtoNet (Snell et al., 2017) is another few-shot classification baseline, where the few-shot samples are averaged and used as a class prototype. We also compare ours with another memory-based classification model, Retrieval-Augmented Classification (RAC) (Long et al., 2022). RAC first retrieves the nearest images from an image memory and feeds their corresponding class text labels to the subsequent text encoder to obtain an auxiliary textual feature, which is then added to the input image feature. RAC was originally designed to be trained with abundant training data for long-tailed classification (Huang et al., 2016). We adapt RAC for few-shot classification and keep the text encoder frozen; otherwise, few-shot training fails to converge. All methods use the CLIP-B/32 backbone.

Results on few-shot classification: Table 8 compares MML and the aforementioned baselines on few-shot classification. While our MML outperforms the other methods, we observe that the performance gap between MML and the linear prob CLIP is bigger with fewer shots. This result implies that *the knowledge retrieval from external memory is especially effective when limited supervised data are available* as the external memory access can compensate for the lack of supervised data.

Extended results on many-shot classification: In addition, we attempt to increase the few shots from the target classes to many shots and demonstrate the performance trend in Table 9. While the linear prob (a) is directly trained with 4 to 256 shots from the target classes, our method (b) is trained on the non-target classes and tested *without* additional training with the 4 to 256 shots. MML outperforms linear prob by a significant margin with 4 shots, and the gain gradually diminishes with increasing training data for linear prob. The results of MML with the test-time training (c) with the 4 to 256 shots show that our model recovers the diminished gap and further improves performance. Note that this work primarily focuses on leveraging external knowledge with zero or minimal supervision, in addition, we also show that additional test-time training benefits MML orthogonally to retrieval-based reasoning.

**Class-incremental classification**

Problem setup: A class-incremental learning model is assumed to receive a set of new class data sequentially and is asked to classify a test image into the accumulated classes. As the model is not assumed to access to the previously seen data, the key challenge is not to forget the old classes.

Details on reproduction: For a fair comparison, we reproduce existing class-incremental learning methods (Li & Hoiem, 2017; Ratcliff, 1990; Rebuffi et al., 2017; Wang et al., 2022a) with the CLIP-B/32 backbone as well as ours on a unified codebase (Zhou et al., 2023a) by following the standard constraints.

Benchmark: We adopt a public benchmark, ImageNet100-Base0-Inc10 (Rebuffi et al., 2017), where 10 unseen classes and their annotated samples are sequentially given for 10 consecutive stages. For each stage, a model is required to classify an image into all the known classes, resulting in the accumulation of 100-class

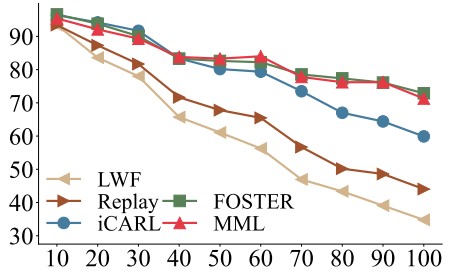

Figure 5: Class-incremental classification on ImageNet100

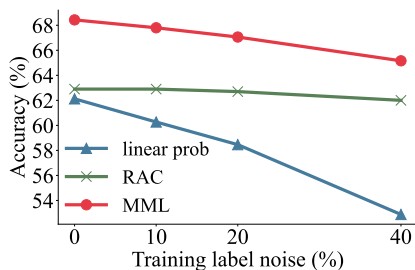

Figure 6: Result with training label noise

classification at the end. Across all stages, the size of the memory for each stage is always restricted to 2000 elements for all methods. For each stage, MML manages the memory length by dropping some old-class data from the memory and storing the new-class data such that the remaining memory elements are closest to the average of the memory contents, following (Rebuffi et al., 2017). Accordingly, MML updates the class prototypes with the updated memory elements at each stage. For evaluation, input images are classified into all the seen classes without stage-specific information.

Results: As seen in Figure 5, MML outperforms or performs on par with the class-incremental learning specialist models, without using specific techniques for the task such as distillation of old class knowledge in model weights (Rebuffi et al., 2017) or storing the heavy model weights to the model memory (Wang et al., 2022a). For CIFAR100, please see Figure 7 in Appendix.

**Training label noise robustness**

We showcase that the reasoning procedure via memory retrieval is robust against the training data label noise. To simulate the label noise, we randomly permute from 10% to 40% of the class labels of training queries with a wrong class and train the architecture with the corrupted labels. This comparison validates the effectiveness of reasoning sources for classification: reasoning from the relevant external knowledge *vs.* reasoning from the memorized parameters. Figure 6 presents the comparison of the baselines and ours on ImageNet1K with the increasing portion of incorrect class labels. The memory-based models, RAC and MML, show robustness and powerful performance against training data noise. As MML predicts classes assisted by retrieving input-adaptive $k$NN from the frozen memory, MML can avoid directly compiling the wrong training data into parameters, particularly being more robust as the more incorrect label noise is injected in training. We hypothesize that retrieval-based reasoning encourages robust learning against the training label noise as the $k$NNs provide interactive reasoning with the neighborhood embeddings.

## 4.5 Feasibility on real-world scenarios

MML is lightweight and introduces little computational overhead; feature extraction, $k$NN retrieval, and knowledge integration take 1129.9, 58.3, and 10.7 GFLOPs (94.2, 4.9, and 0.9 %), respectively. MML is efficient in that training with batch size 256 consumes only 2.2 GB GPU memory on a 2080Ti thanks to the frozen backbone and memory features. We verify that classification inference with MML scales up to 1000 classes at once which consumes only 3.4 GB memory on a single GPU. As MML is scalable with the increasing number of classes with the manageable size of memory, MML is expected to handle the dynamic number of classes for classification tasks in the real world.

The frozen pre-trained encoders significantly contribute to the little computation overhead and are considered the prerequisite of MML. Conversely, MML is implausible to be trained without such pre-trained encoders. We have attempted to train MML from scratch and achieved nearly random accuracies of 2.4% and 5.8% on ImageNet-S and CUB, respectively. This is due to the unreliable contents in the memory and the lack of training data to train image-text encoders. This reliance of pre-trained encoder might hinder application on specialized target domains such as medical, industrial vision, or domains where CLIP is not applicable. Such domains necessitate specialized image-text encoders pre-trained on each domain to aim higher precision.

# 5 Conclusion

We have presented the memory-modular learner and demonstrated its efficacy in various scenarios, investigating the memory-modular generalization for unseen classes. The experiments show that our memory-modular reasoning effortlessly generalizes to unseen classes with memory replacement and exhibits robustness to noisy memory data. We also frame our retrieval-based zero-shot classification as web-assisted zero-shot classification, which is believed to be more realistic in the future research with the growth of web-trained foundation models. We believe that memory-modular learning benefits various tasks in the areas of artificial intelligence beyond classification, leaving them for future work.

**Acknowledgments**

This work was supported by Samsung Electronics Co., Ltd. (IO201208-07822-01) and the IITP grants (2022-0-00290: Visual Intelligence for Space-time Understanding and Generation based on Multi- Layered Visual Common Sense (50%), 2022-0-00113: Sustainable Collaborative Multi-modal Lifelong Learning (50%)) funded by Ministry of Science and ICT, Korea.

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

Table 10: Class split and the numbers of memory items per class of the datasets used in this paper

| dataset | number of classes train/val/test | avg. memory items per class image | avg. memory items per class text |
|---|---|---|---|
| *datasets for single-dataset zero-shot transfer* | | | |
| ImageNet-S (Russakovsky et al., 2015) | 600/200/200 | 760.0 | 209.0 |
| CUB (Wah et al., 2011; Akata et al., 2013) | 150/50 | 65.1 | 147.6 |
| *datasets for cross-dataset zero-shot transfer* | | | |
| ImageNet1K (Russakovsky et al., 2015) | 1000 | 760.0 | 209.0 |
| Caltech101 (Fei-Fei et al., 2004) | 100 | 73.2 | 292.0 |
| OxfordPets (Parkhi et al., 2012) | 37 | 70.6 | 240.1 |
| Cars196 (Krause et al., 2013) | 196 | 80.9 | 219.0 |
| Flowers102 (Nilsback & Zisserman, 2008) | 102 | 82.7 | 153.2 |
| Food101 (Bossard et al., 2014) | 101 | 64.7 | 157.3 |
| FgvcAircraft (Maji et al., 2013) | 100 | 76.7 | 295.2 |
| SUN397 (Xiao et al., 2016) | 397 | 75.4 | 209.9 |
| DTD (Cimpoi et al., 2014) | 47 | 71.1 | 169.5 |
| EuroSAT (Helber et al., 2019) | 10 | 70.4 | 208.1 |
| UCF101 (Soomro et al., 2012) | 101 | 74.9 | 226.2 |

# A  Appendix

In this appendix, we provide additional experimental details and results of our method. We will make our attached code and data publicly available once accepted.

## A.1  Clarification of the term "zero-shot" classification using web data

Here we continue the discussion with the zero-shot classification from Sec. 2 and justify the task of web-assisted zero-shot classification. In our "zero-shot" classification experiments, MML uses the web-crawled images in memory, which are collected by web search with the class names in text and thus are inevitably noisy. The use of such images may appear to mismatch with the term "zero-shot". However, note that we use the term "shot" to refer to *human-annotated* oracle images as conventionally used in the zero-/few-shot learning literature (Xian et al., 2017; Wang et al., 2020; Zhai et al., 2022; Liu et al., 2023). In this context, MML is indeed a *zero-shot* learner assisted with web data, which takes advantage of external memory data in contrast to conventional zero-shot methods without it. This zero-shot approach generalizes to arbitrary classes by replacing memory without training and shows strong robustness to noisy memory contents (*cf.*, Table 6).

## A.2  Additional implementation details

All input images are normalized and resized to $224 \times 224$ before being fed to the CLIP encoder following the official implementation. [3] All texts are truncated with 75 words before being fed to the CLIP text encoder. For image and text $k$NN attention layers, we use a single layer for each image and text branch and implement with the transformer encoder implementation of CLIP. Please refer to the attached code for the precise implementation details.

## A.3  Details on web-crawled datasets

Table 10 shows the composition of the datasets we used for experiments. For the image memory for ImageNet benchmarks, we use a publicly available dataset, WebVisionV1 (Li et al., 2017), which consists of 1,000 class images crawled from Google and Flickr. For the other datasets, there are no such web-crawled image datasets of corresponding to the classes of each dataset. We thus actually crawl images by using an automatic image

---

[3]https://github.com/openai/CLIP

Table 11: Cross-dataset zero-shot transfer results with and without 6 duplicated images in the memory with the test set

| MML | 1 shot | 4 shots |
|---|---|---|
| test images containing duplicates with memory | 62.80 | 66.20 |
| test images *not* containing duplicates with memory | 62.79 | 66.19 |

Table 12: Comparison of retrieval-based classifiers on ImageNet-S with varying length of memory

| # mem elements per class | 1 | 10 | 100 | 500 | full ($\approx 760$) |
|---|---|---|---|---|---|
| $k$NN classifier (Nakata et al., 2022) | 55.3 | 66.9 | 73.6 | 76.5 | 76.4 |
| MML | **80.4** | **82.0** | **83.0** | **83.1** | **83.0** |

Table 13: Comparison of retrieval-based classifiers on ImageNet-S with different values of $k$ in $k$NN

| $k$ of $k$NN | 8 | 16 | 32 | 64 |
|---|---|---|---|---|
| $k$NN classifier (Nakata et al., 2022) | 75.2 | 76.3 | 76.4 | 76.0 |
| MML | **83.0** | **83.1** | **83.0** | **82.7** |

crawler [4] that searches a text keyword on Google and downloads the images. For text memory, we search the class text name on English Wikipedia [5] using a Wikipedia crawler [6] to retrieve the texts of the article, where an example is shown in Table 18. We also add 80 text phrases such as "a photo of [class]", which is provided by CLIP as done in Tian et al. (2022). Each sentence in the retrieved articles corresponds to an item in the text memory. In total the crawled images amount to 89,970 images for 11 datasets and 465,496 text sentences from 2,141 Wikipedia articles for text memory of 12 datasets. We inspect that 6 images among all the web-crawled images turn out to be the same of the test images. We thus remove the 6 images from the memory and evaluate the model in the exact same setup with that of Table 1. As shown in Table 11, the accuracy is not affected by the negligible amount of duplicated images from the test set.

## A.4 Additional experimental results

### Effect of memory length

Table 12 shows the numerical results of Figure 3 in the main manuscript.

### Effect of $k$ in $k$NN

We vary $k$ for the two retrieval-based models and compare their accuracy in Table 13. Both the retrieval-based models reach sweet spots at a certain $k$, in this example at around 16, and continue to drop with $k \geq 32$ perhaps because more irrelevant $k$NNs negatively affect attentive integration. Note that MML outperforms the $k$NN classifier with all difference choices of $k$.

### MML with another image-text encoder backbone

Table 14 compares the baselines and MML that use ALIGN (Jia et al., 2021a) as an alternative image-text encoder of CLIP. A baseline "zero-shot ALIGN" selects the class of the highest text similarity with class text labels in text prompts such as "a photo of [class]". ALIGN is trained with a web-crawled image and text pairs as CLIP but on a slightly heavier image and text encoder (EfficientNet-L2 (Tan & Le, 2019) and BERT-Large (Kenton & Toutanova, 2019)), which amount to 172M parameters in total, compared to 151M parameters of CLIP. As the official model checkpoints are not available, we adopt the released checkpoints and the data preprocessor provided by a third party. [7] In this experiment, except for the backbone, all the experimental settings remain the same as those in Zhou et al. (2023a). We observe that MML exhibits more outstanding performance than baselines and show that MML is not specific to a certain image-text encoder.

---

[4] https://github.com/YoongiKim/AutoCrawler
[5] https://en.wikipedia.org/wiki/Main_Page
[6] https://github.com/goldsmith/Wikipedia
[7] https://huggingface.co/kakaobrain/align-base

Table 14: Cross-dataset zero-shot transfer with ALIGN (Jia et al., 2021a) image-text encoder. MML is learned with 1 to 4 samples from 1000 ImageNet1K classes and transferred to 10 other datasets with zero shot.

| method | ImgNet1K | Caltech101 objects | OxfordPets pets | Cars cars | Flowers flowers | Food food | Aircraft airplanes | SUN scenes | DTD textures | EuroSAT land | UCF actions | avg. |
|---|---|---|---|---|---|---|---|---|---|---|---|---|
| zero-shot ALIGN | 65.9 | 79.7 | 62.5 | 69.5 | 53.0 | 76.1 | 8.3 | 44.5 | **51.4** | 23.4 | 64.1 | 54.4 |
| $k$NN classifier (Nakata et al., 2022) | 62.2 | 87.0 | 68.6 | 77.5 | 67.8 | 71.8 | **24.2** | 59.1 | 37.9 | **28.2** | 61.3 | 58.7 |
| MML (ImageNet-1) | 67.2 | **94.2** | 76.1 | 77.3 | **67.9** | 77.7 | 22.0 | 67.2 | 47.3 | 23.8 | 66.1 | 62.4 |
| MML (ImageNet-4) | **69.1** | 93.3 | **76.2** | **77.9** | 66.7 | **78.2** | 21.6 | **67.9** | 49.5 | 24.8 | **66.3** | **62.9** |

Table 15: Single-dataset zero-shot classification with CLIP (Radford et al., 2021) and ALIGN (Jia et al., 2021a) image-text encoders.

| | ImageNet-S | | | CUB | | |
|---|---|---|---|---|---|---|
| backbone | CLIP B/32 | CLIP L/14 | ALIGN base | CLIP B/32 | CLIP L/14 | ALIGN base |
| zero-shot CLIP/ALIGN | 82.8 | 90.1 | 85.1 | 61.9 | 73.8 | 55.9 |
| $k$NN classifier (Nakata et al., 2022) | 76.4 | 86.5 | 82.4 | 67.9 | 82.6 | 64.1 |
| MML | **83.0** | **91.1** | **86.1** | **75.6** | **87.8** | **73.7** |

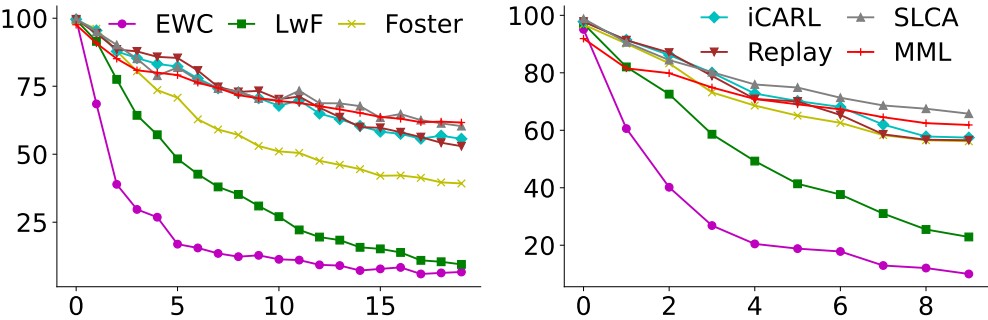

Figure 7: Class-incremental classification on CIFAR100 Base5-inc5 (left), Base10-inc10 (right)

**Class incremental classification on CIFAR100**

We conduct experiments on CIFAR100 and compare MML with the other methods dedicated to class-incremental classification in Fig. 7. To reproduce the class-incremental learning baselines with CLIP, we adopt the unified code base from a survey paper (Zhou et al., 2023a) and replace the backbone from ResNet18 to CLIP-B/32. Note that MML cannot be implemented with ResNet18 as CLIP-ResNet18 is unavailable. As the backbone is switched, we tune learning rates and epochs for each method and present their best results for a fair comparison. The results show that MML performs comparably to the class-incremental learning models on CIFAR100 as well, but achieves less gain compared to the performance on ImageNet100. We hypothesize that this is due to the majority of web-crawled images in the wild displaying a distribution that is markedly different from the $32 \times 32$ object-centric images in CIFAR.

**Few-shot classification on mini-ImageNet**

In Table 16, we provide an additional experimental result using CLIP-B/32 on a public few-shot image classification benchmark: mini-ImageNet (Vinyals et al., 2016). We also include the two existing methods evaluated on the benchmark (He et al., 2022; Hu et al., 2022), which leverage a vision foundation model via fine-tuning it. The seminal few-shot classification work based on deep learning (Vinyals et al., 2016) proposes this benchmark, thus it configures a toy experimental setup in the nowadays perspective. Derived from ImageNet (Russakovsky et al., 2015), mini-ImageNet consists of $84 \times 84$ sized downsampled images from 64/16/20 object classes for train/val/test splits, respectively. Compared to ImageNet-S with 600/200/200

Table 16: Comparison on few-shot classification

| # target classes
# shot | mini-ImageNet
20 classes
1 shot | ImageNet-S
200 classes
4 shots | 

16 shots |
|---|---|---|---|
| linear-prob CLIP (Radford et al., 2021) | 61.3 | 72.1 | 80.6 |
| He et al. (2022) | 74.7 | - | - |
| Hu et al. (2022) | 95.3 | - | - |
| RAC (Long et al., 2022) | 69.1 | 66.8 | 78.1 |
| kNN classifier (Nakata et al., 2022) | **96.6** | 77.2 | 77.2 |
| MML (ours) | **96.6** | **82.8** | **83.5** |

Table 17: Single-dataset zero-shot transfer on ImageNet-S with varying the temperature hyperparameter, $\tau$. CLIP-B/32 is used.

| | 1 | 4 | 8 | 16 | 32 | 64 |
|---|---|---|---|---|---|---|
| MML | 80.0 | 82.3 | 83.5 | **83.6** | 83.5 | 82.9 |

class split with the original image size, the benchmark is less challenging in terms of the smaller image size and the easier categorization difficulty. As shown in Table 16, kNN classifier and MML show comparable performance on mini-ImageNet probably because the easier problem setup makes kNN classifier relatively powerful than the method involving learning. However, when it comes to the more complected benchmark, kNN classifier is no longer powerful as MML on ImageNet-S. The memory-modular approach, which meta-learns to generalize unseen classes, is shown substantially more helpful on the more complex real-world classification scenarios than a toy few-shot classification problem.

**Supervised image classification**

We also signify the efficacy of MML on the standard supervised image classification in Table 19. MML can also tackle this classification by setting the memory content and the prototype *unchanged* between training and testing to handle the known closed-set target classes.

Problem setup: In contrast to the unseen-class classification of the previous paragraphs, the more primitive definition of image classification assumes that the target classes are identical across training, validation, and testing. In other words, the generalization beyond the seen classes during training is not the focus of this classification task.

Baseline: Note that there exist no prior zero-shot methods that use dynamic memory, directly comparable to ours. A similar method with external memory retrieval, RAC (Long et al., 2022), is compared. RAC tackles seen class problems, whereas MML focuses on generalizing beyond seen classes with memory replacement.

Benchmark: The 11 datasets in Table 1 are used. For each dataset, 4 random images per class are used for training. All methods are evaluated on each dataset using the CLIP-RN50 backbone.

Results: The results are shown in Table 19. MML performs more accurately than other baselines on average. In particular, our model demonstrates greater effectiveness when supplemented with external web-crawled data that provide relevant features for classification. For instance, the diverse viewpoints and color variations of car images on the internet benefit for car model categorization (Cars196). Overall, the proposed MML signifies its effectiveness also on the standard supervised and fine-grained image classification.

**Logit temperature hyperparameter $\tau$.**

Table 17 demonstrates the effect of the temperature hyperparameter $\tau$ of Eq. 5. We empirically observe that adjusting the smoothness of the logit plays an important role in effective training. Note that a higher value for $\tau$ encourages the logit value more indistinguishable to each other, *i.e.*, smoothing effect. We observe that the loss for training hardly converges with $\tau = 1$ as the loss is not big enough to penalize the model. We thus increase the temperature value from 1 to 64 and notice the trend; the performance reaches the highest peak at a certain point, *e.g.*, around 16 on ImageNet-S, and continues to drop afterward. This experiment shows that $\tau$ controls the degree of the logit smoothness and is a crucial hyperparameter for effective training.

Table 18: The first five sentences from the Wikipedia article of the "tench" class

The tench or doctor fish (Tinca tinca) is a fresh- and brackish-water fish of the order Cypriniformes found throughout Eurasia from Western Europe including the British Isles east into Asia as far as the Ob and Yenisei Rivers. It is also found in Lake Baikal. It normally inhabits slow-moving freshwater habitats, particularly lakes and lowland rivers. The tench was formerly classified in the subfamily Leuciscinae with other Eurasian minnows, but more recent phylogenetic studies have supported it belonging to its own family Tincidae. The tench is most often found in still waters with a clay or muddy substrate and abundant vegetation. This species is rare in clear waters across stony substrate, and is absent altogether from fast-flowing streams. It tolerates water with a low oxygen concentration, being found in waters where even the carp cannot survive. ...

Table 19: Supervised classification results trained with 4 shots from the target (seen) classes from each dataset

| method | ImgNet1K | Caltech101 | OxfordPets | Cars | Flowers | Food | Aircraft | SUN | DTD | EuroSAT | UCF | avg. |
|---|---|---|---|---|---|---|---|---|---|---|---|---|
| zero-shot CLIP (Radford et al., 2021) | 58.2 | 86.3 | **85.8** | 55.6 | 66.1 | **77.3** | 17.3 | 58.5 | 42.3 | 37.6 | 61.5 | 58.8 |
| linear prob (Radford et al., 2021) | 41.3 | 84.3 | 56.4 | 48.4 | **84.8** | 55.2 | **23.6** | 54.6 | 50.1 | **68.3** | 62.2 | 57.2 |
| ProtoNet (Snell et al., 2017) | 38.3 | 81.6 | 59.0 | 45.9 | 81.3 | 52.2 | 21.6 | 54.0 | 48.0 | 65.3 | 64.0 | 55.6 |
| kNN classifier (Nakata et al., 2022) | 55.7 | 79.7 | 59.9 | 55.8 | 65.6 | 60.4 | 20.5 | 50.2 | 29.0 | 19.4 | 45.2 | 49.2 |
| RAC (Long et al., 2022) | 37.0 | 83.6 | 69.4 | 52.9 | 78.9 | 59.4 | 20.6 | 53.0 | 49.8 | 57.4 | 59.0 | 56.5 |
| MML | **66.2** | **90.2** | 85.5 | **64.8** | 84.1 | 76.7 | 21.3 | **65.8** | **52.4** | 44.1 | **70.1** | **65.6** |

Table 20: Performance with the increasing noise level in the memory to construct class prototypes on ImageNet-S

| noise level | 20 % | 10 % | 5 % | 0 % |
|---|---|---|---|---|
| avg of random instances | 79.7 | 80.8 | 80.6 | 83.0 |
| avg of highest cross-modal consensus (ours) | 83.0 | 82.7 | 82.7 | 83.0 |

Table 21: Positive and negative similarity statistics of the prototype classifier and MML

| | prototype classifier | MML |
|---|---|---|
| classification accuracy | 80.1 | **83.0** |
| avg positive class similarity | 0.654 | 0.388 |
| avg negative classes similarity | 0.361 | 0.090 |
| positive - negative similarity gap | 0.293 | **0.298** |

**Robust performance with prototype construction from noisy data pool**

Table 20 shows the robustness of the proposed cross-modal consensus prototype generation method to the data noise. As described in Section 3.1, the class prototypes are built by averaging the memory items that have the highest cross-modal similarity of the same class from the other modality memory. This prototype construction effectively ignores the noise in the memory by leveraging the cross-modal memory consensus.

Experimental setup: We experiment with two copies of the memory. For one memory to construct zero-shot class prototypes, we randomly shuffle the memory items with other classes such that a certain percentage of memory items are from the web-crawled instances of other class names. For the other memory for kNN retrieval being fed to the subsequent knowledge integration, the memory is set unchanged.

Results: The consensus-based class prototype construction is shown to be robust to the data noise. Even with the 20 % of mislabeled memory items, the class prototypes are constructed as accurately as the clean prototype within the error bound. If the prototypes are the average of random memory instances of each class, the performance drops with the increasing noise level. However, the proposed consensus-based prototype construction is able to ignore the unrelated noise as it uses only the most similar image memory items to the text memory, and from image memory to the text prototypes also.

**Analysis of class prototype similarity**

Table 21 provides the average distance between all evaluation images and the prototypes of each model. This experiment investigates the behavior of the learnable integration of MML: how it updates the input image embedding to be close or distant to the class prototypes. Note that the class prototypes are static, *i.e.*, both

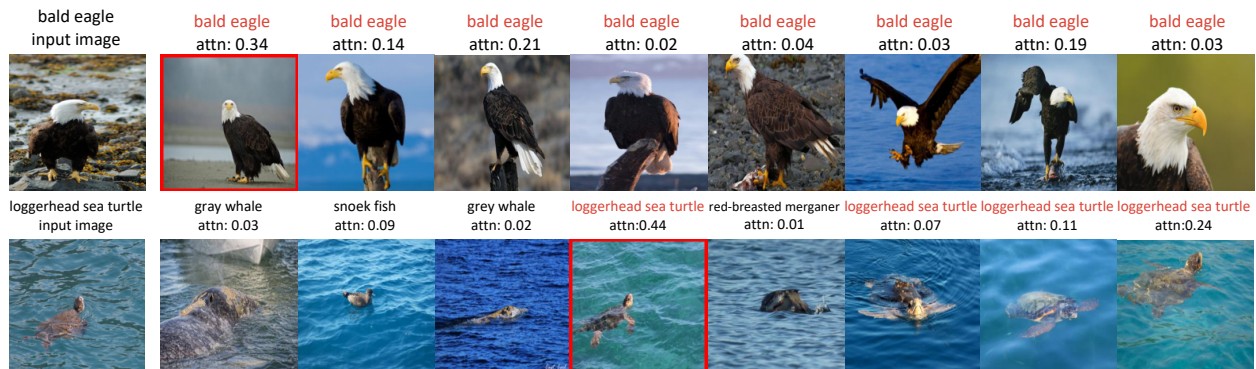

Figure 8: An input query (leftmost) from ImageNet and its 8NN images retrieved from the WebVision memory using CLIP ViT-B/32. "attn" denotes the attention weight value, and the red frame denotes the image which gained the highest attention among 8NNs.

model uses exactly the same class prototypes, but MML updates the embedding through the knowledge integration layer. We observe that MML pushes the embedding to be far away especially from the negative class prototypes, achieving the strong final classification result.

**Visualizations of retrieved $k$NNs.**

Figure 8 provides examples of the image retrieval results. The leftmost image is the input query from ImageNet, and its 8NNs retrieved from the WebVision image memory are presented on the right. Note that the retrieved 8NNs exhibit the superficially similar appearance of the query but often belong to different classes, *e.g.*, turtle floating on the sea. The subsequent attentive knowledge integration process then reweights the NNs with soft attention weight values. In the second example, the 1NN appears similar to the query but is a different class, however, its attention value is down-weighted and contributes insignificantly to the attentive aggregation. The attentive aggregation meta-learns to function independently of memory contents and is able to perform effectively with unseen memory contents. We also present the assigned attention weight values in attentive knowledge aggregation, which are the similarity of the query and the NNs after **Q** and **K** projection in Eq. 3. The attention weight value is translated as the learned aggregation weight for the NNs *to what extent each element is contributed for aggregation* among the NNs to represent the query. The attentive knowledge integration process reweights the NN elements with soft attention weight values based on the attentive convex-combination similarity such that the semantically relevant NN elements are more attended.

