# OpenReview forum: "Memory-Modular Classification: Learning to Generalize with Memory Replacement"
_TMLR — Accepted by TMLR_

### Review · Reviewer_9m1F · 2024-11-20

**Summary Of Contributions:**

The paper proposes a memory-modular learner (MML) for image classification that separates knowledge storage from model reasoning. The key innovation is using modular external memory that can be updated/replaced without model retraining. The memory consists of web-crawled images and text data collected through keyword searches. The model uses cross-attention to integrate memory with input and prototype-based classification.

**Audience:**

Yes

**Claims And Evidence:**

Yes

**Requested Changes:**

(1) emphasize the specific technical differences from REVEAL (Hu, et al. CVPR2023), especially the design of the framework such as the use of K-NN instead of learnable modules for retrieval or so.

(2) The experiments need more details to ensure the fairness of comparisons, and also include more details when conducting on different image classification problem setups.

**Strengths And Weaknesses:**

Strength: (1) A new image classification framework using replaceable memory. (2) Comprehensive evaluation across multiple downstream image classification tasks. (3) The experiments show promising performance with robustness to noisy web data, and computational efficiency in terms of memory usage.

Weakness: (1) While the work targets the image classification task, the framework is still similar to in REVEAL (Hu, et al. CVPR2023), it would be better to put more emphasis on the difference and also which proposed component really makes this framework unique and effective. (2) There is limited discussion about the size of external memory given the concern when deploying it to very large real-world tasks. (3) The experiments should include additional ablation studies to validate the use of some key components in the framework such as the use of cross-attention for knowledge integration and the NN retrieval. (4) The comparison with different downstream tasks could be unfair as the proposed method use web-crawled data while the baseline use curated data only. In addition, for class-incremental learning, did all the compared methods use the pre-trained CLIP model or not?

---

> ### Author Response · Authors · 2025-01-09
> **We have updated the manuscript following the suggestions and marked the modified parts in red.**
>
> Thank you for your comments. We have updated the manuscript following the suggestions and marked the modified part in red. The changes are found in Sections 2.2, 4.3, 4.4, 4.5. We additionally leave a message to summarize or supplement the change in the manuscript.
>
> 1. Weakness (1): The difference between REVEAL (Hu, et al. CVPR2023).
> * The memory-augmented learning systems of REVEAL (Hu et al. CVPR 2023) and MML are indeed similar in terms of architecture. But they are different in terms of the role of _specific_ (MML) vs _general_ (REVEAL) knowledge memory.
> * The difference is now elaborated in **Section 2.2**.
>
> 2. Weakness (2): Limited discussion about the size of memory.
> * We have modified the subsection title and appended the following discussion point in **Section 4.4 Discussion on scalability**.
> * We verify that classification with MML scales up to 1000 different classes at once which consumes only 3.4 GB memory on a single GPU. As MML is scalable with the increasing number of classes with the manageable size of memory, MML is expected to effortlessly handle the dynamic number of classes for classification tasks in the real world.
>
> 3. Weakness (3): Ablation study on key components
> * The old manuscript indeed included the validation of key components but spread out across multiple tables of different benchmarks. We have gathered the experimental data and neatly merged them into a new **Table 3** that ablates the two key components of MML. We also inserted a new paragraph that analyzes the experiment in the paragraph **Ablation study on model components** in **Section 4.3 Model analyses**.
>
> * In the old manuscript, the usage of cross-attention is verified in Table 1 and Table 15. The kNN classifier corresponds to the NN retrieval model without the learnable cross-attention part. Table 1 contains the comparison of kNN classifier and MML on 11 datasets. Table 15 contains the same comparison on ImageNet-S and CUB, which is now moved to the row (b) of the new **Table 3**.
> * Next, the use of NN retrieval was verified in the Table 5 with the first and the last rows. The no kNN baseline uses the attention block without retrieval, self-attending the input feature without kNNs. This result is moved to the row (c) of the new **Table 3**.
>
> 4. Weakness (4): Fairness on comparison. For class-incremental learning, did all the compared methods use the pre-trained CLIP model or not?
> * Yes, and the detail is included in the original manuscript. The beginning of the ``Class incremental classification’’ paragraph in Section 4.4 explains the experimental details which states that
> > we reproduce existing class-incremental learning methods (Li & Hoiem, 2017; Ratcli, 1990; Rebu et al., 2017; Wang et al., 2022a) with the CLIP-B/32 backbone as well as ours on a unified codebase (Zhou et al., 2023a) by following the standard constraints.
> * However, we have improved the visibility of the experimental details by adding the paragraph header.
>
> 5. Requested change (1): Please refer to Weakness (1)
>
> 6. Requested change (2): More details required to ensure the fairness of comparisons
> * We have clarified the implementation detail parts with the paragraph headers. Also, we have elaborated the problem setup and the reproduction details for the different classification problem experiments in **Section 4.4**.

---

### Review · Reviewer_7Wbr · 2024-12-09

**Summary Of Contributions:**

The paper presents a novel architecture, Memory-Modular Learner (MML), designed for image classification tasks. MML introduces the idea of external, replaceable memory populated by web-crawled image and text data to enable generalization to unseen classes without retraining. It demonstrates applications in zero-shot, few-shot, fine-grained, and class-incremental classification tasks.

**Audience:**

Yes

**Broader Impact Concerns:**

The method provides the opportunity to easily adapt a pre-trained foundation model to different downstream tasks, which is an important research direction for the present time. The solution holds the potential to revolutionize the foundation model adaptation literature.

**Claims And Evidence:**

Yes

**Requested Changes:**

-  While the authors demonstrate robustness to noisy data, they could provide more detailed analyses on the quality of web-crawled memory contents. For instance, how do different noise levels or retrieval methods affect the generated prototypes?
- Addressing how MML handles domain shifts or misaligned retrievals would be valuable. Could the system incorporate mechanisms to detect or filter irrelevant memory items dynamically?
- While the modular memory concept is compelling, the paper could include more detailed ablations to assess how memory size and content diversity impact performance.

**Strengths And Weaknesses:**

**Strengths:**

- Innovative Use of Modular Memory: The concept of leveraging replaceable external memory for flexible adaptation to unseen tasks is innovative and potentially impactful.

- Robust Experimental Results: The authors provide a wide range of experiments to validate the model's performance across different tasks, showcasing robustness against noisy data and generalization to unseen classes.

- Efficiency Considerations: The paper addresses computational overhead effectively, showing the method's scalability and lightweight implementation.


**Weaknesses:**

1. Introduction Lacks Method Details:
The introduction outlines the motivation and general approach of MML but fails to discuss its specific workings. Readers unfamiliar with the concept may struggle to connect the proposed framework with the results presented later. For example, the introduction should briefly touch on key ideas such as prototype-based classification and the retrieval-augmented learning process.

2. Insufficient Related Work on Class-Incremental Learning:
The discussion on class-incremental learning (CIL) is incomplete and somewhat misaligned with prior research. Although the authors differentiate MML from CIL approaches reliant on distillation, they overlook works that similarly incorporate external memory or modular components. A more comprehensive exploration of CIL strategies and their limitations would strengthen the paper.

3. Overreliance on External Factors:
Web-based Data Collection: MML's dependence on web-crawled data introduces potential biases and limitations. The reliance on keyword-based retrieval may result in noisy or irrelevant samples, especially when the search engine model has domain-specific calibration issues.
Zero-shot Class Prototypes: The method generates class prototypes using zero-shot predictions, which are inherently noisy. This reliance may propagate errors, particularly in domains where text-image alignment is miscalibrated.
4. Model Dependency on Pre-trained Encoders:
The proposed architecture is heavily reliant on pre-trained encoders like CLIP for both memory construction and feature extraction. This dependency raises concerns about the generality of the approach if applied to domains not well-supported by such models.

---

> ### Author Response · Authors · 2025-01-09
> **We have updated the manuscript following the suggestions and marked the modified parts in green.**
>
> Thank you for your comments. We have updated the manuscript following the suggestions and marked the modified part in green. The changes are found in Sections 1, 2.3, 4.3, 5, A.4. We additionally leave a message to summarize or supplement the change in the manuscript.
>
> 1. Weakness (1): Introduction Lacks Method Details
> * We have elaborated the main methodological keywords of prototype-based classification and retrieval-augmented learning process in **Section 1**.
>
> 2. Weakness (2): Insufficient exploration of related work on class-incremental learning.
> * The data memory is both used in MML and some work of CIL, however, their purposes are different. The memory of CIL helps not to forget the previously seen classes, however, the memory of MML assists the current classes of interest, which might have not been seen.
> * We have clarified the similarity and difference between CIL and MML in **Section 2.3**.
> * We have also cited other CIL methods that also similarly use the memory system.
>
> 3. Weakness (3): Overreliance on External Factors: Web-based data collection and zero-shot class prototypes
> * It is true that MML leverages web-crawled data, which is impossible to completely remove the noise.
> Thus we design MML to minimize the noise and balance the trade-off between avoiding additional human annotation and data noise.
> The model design effort is validated by the following experiments that MML is robust to noise.
> * **Table 4** shows that using all or random memory items for prototype construction degrades classification, which validates the effectiveness of the retrieval-based prototype generation of Eq. (1).
> * **Table 7** compares the human-annotated and the web-crawled data memory, where the zero-shot web-crawled memory performs on par with the upperbound of the human-annotated memory.
>
> 4. Weakness (4): Model Dependency on Pre-trained Encoders
> * The use of pre-trained generalist encoders introduces the trade-off between general and specialized recognition. This following discussion point is added to **Section 5. Conclusion**.
> * We choose to use the pre-trained generalist encoders as _the goal of MML is to generalize to new classes_ by simply replacing the memory contents without model retraining. MML is shown to generalize well on unseen fine-grained classes with zero shot (**Table 1**).
> * However, highly specialized domains, e.g., medical or industrial vision, would necessitate specialized image-text encoder for higher precision.
>
> 5. Requested Changes (1): More analyses on the quality of memory contents about different noise levels
> * Thank you for the great suggestion. We have experimented with the increasing noise levels in memory and showed our *cross-modal consensus-based prototype construction is robust to data noise*.
> * The experimental results and analyses are shown in the new **Table 20** in the supplementary material.
>
> 6. Requested Changes (2): Domain shift: Could the system incorporate mechanisms to detect or filter irrelevant memory items dynamically?
> * Yes. MML robustly handles various domain-shift scenarios as shown in **Table 1**, but we miss to explain it with the keyword "domain shift". We have added the keyword "domain shift" in the table caption in Table 1 for visibility.
> * Table 1 signifies that the MML handles the domain-shift scenarios where the evaluation classes, memory items, and evaluation images (_fine-grained diverse classes_) completely diverge from the ones from training  (_general objects_) .
>
> 7. Requested Changes (3): Assessment of how memory size and content diversity impact performance.
> * The effect of size is examined in **Table 12** in the supplementary material. However we agree that this experiment is crucial, thus we have plotted the results in the **new Figure 3** in the new manuscript. Increasing the number of memory items is beneficial for classification, however, the benefit converges with the sufficiently large number of memory items per class.
>
> * The diversity level of memory contents are examined in Tables 6 and 7.
> **Table 6** shows the memory contents comparison with different crawling sources. MML shows the marginal performance difference within errorbound across different memory contents.
> * **Table 7** compares the noiseness level of the memory contents with human-annotated memory and unsupervised web-crawled memory with ImageNet1K and WebVision. Our MML method based on the web-crawled memory (WebVision) already outperforms the model without memory and is on par with the method with the clean human-annotated memory (ImageNet1K).

---

### Review · Reviewer_1vVW · 2024-12-27

**Summary Of Contributions:**

The paper studies a multi-modality in terms of input data where each datum is a pair of an image and its text description to perform classification. In particular, the paper proposes a modular memory based method that can modify or replace its modular memory without re-training the whole model to perform classification and zero-shot learning. The main idea is to employ k-nearest neighbours to retrieve some samples in each target classes to integrate them into the features of the sample of interest before calculating its (the sample of interest) similarity to each prototype for classification. Due to such modularity, the proposed method is quite flexible, especially, it only needs to learn the cross-attention to integrate the nearest neighbours into the (image and text) features of a sample of interest. In terms of empirical performance, the proposed method shows promising results in several benchmarks.

**Audience:**

Yes

**Claims And Evidence:**

No

**Requested Changes:**

In the current form, the paper is quite simple and does not clearly differentiate itself from previous studies, especially compares to Prototypical Networks and CLIP. The main request is to understand more about the differences, explain why additional steps proposed in Eqs. (2) and (3) help to improve classification results, as well as ablation studies when there is no such step involved or when one of those two steps has high errors.

Another request is to minorly revise the section of Related Work, where one relevant study "Infinite Mixture Prototypes for Few-Shot Learning" by Allen et. al presented at ICML 2019.

Some minors to improve the clarity of the paper:
- The abbreviation "NN" is mentioned early in the paper, but its full-form is only explained at page 8.
- The abuse of notations in Eq. (4) should be avoided and each element of the similarity vector z should be used. What I mean is as follows:

$$
z_{c} = \operatorname{cosine\\_sim}(p\_{c}^{\mathrm{img}}, f) +  \operatorname{cosine\\_sim}(p\_{c}^{\mathrm{txt}}, f).
$$

**Strengths And Weaknesses:**

**Strengths**
- The paper investigates an important setting in classification where each data point consists of multiple modalities (here is image-text pair). In classification, such a setting has not been widely studied, and hence, makes the problem significant and interesting. Given the additional modalities, classification performance could be improved further.
- The proposed method in the paper is simple but effective. Its main idea replies on prototype-based classification with an additional step "augmenting" features of a sample of interest before measuring the similarity to each target prototype. Such a simplicity allows the proposed method to quickly adapts to new settings without the need to re-train the whole model.

**Weaknesses**

The expecation after the introduction of the paper is how to fuse multiple-modalities to improve classification performance. However, when seeing Eq. (4) where the fusion happens by simply adding those modality embeddings, there is no "wow", but a slight disappointment.

Furthermore, the paper, in a loose sense, can be considered as an extension of Prototypical Networks (Snell et al. 2017) applied on multi-modality data. The main difference is at "augmenting" features of the sample of interest by using the features of its k-nearest-neighbours as shown in Eqs (2) and (3). Despite its empirical effectiveness, it lacks the explanation or intuition on why such steps improve the results compared to Prototypical Networks.

Another concern is that the proposed method heavily relies on models that are well-trained to extract features. This could be useful in some common settings, such as vision-language models on text and natural images. However, when working on uncommon domains, such as medical or space exploration, where there is no pre-trained models, extracting good features to measure their similarity leads to a severe issue and could break the proposed method. Thus, claiming that there is no need to re-train the whole model might not be exactly correct, especially when using pre-trained ones.

---

> ### Author Response · Authors · 2025-01-09
> **We have updated the manuscript following the suggestions and marked the modified parts in blue.**
>
> Thank you for your comments. We have updated the manuscript following the suggestions and marked the modified part in blue. The changes are found in Sections 3.1, 3.2, 5, A.4. We additionally leave a message to summarize or supplement the change in the manuscript.
>
> 1. Weakness (1): Lack of explanation/intuition/comparison why this work and how to improve ProtoNets
> * The MML architecture is indeed considered as an advanced Prototypical Networks with zero-shot prototypes and feature augmentation with the kNNs of the input. We thus measure our key methodological difference, the feature augmentation effect, in **Table 21** to see how much the kNN integration layer updates the feature embedding. The result shows MML effectively pushes the embedding to be far away, especially from the negative class prototypes, achieving the strong final classification result.
>
> 2. Weakness (2): Heavy reliance on pre-trained features.
> * The use of pre-trained generalist encoders introduces the trade-off between general and specialized recognition. Our model does generalize on unseen fine-grained classes such as flowers and airplanes with zero shot (**Table 1**). However, highly specialized domains, e.g., medical or industrial vision, would necessitate specialized image-text encoder for higher precision.
> * This point is added in **Section 5 Conclusion**.
>
> 3. Requested Changes (1): Differentiation/understanding the proposed method from previous studies, explanation of why additional steps Eqs. (2) and (3) are helpful.
> * We have strengthened the paper argument by reflecting the suggestions.
> * The ablation study from the MML components is summarized in **Table 5**.
> * The difference between ProtoNets are described in **Section 3.1**.
> * The effect and analysis of the proposed knowledge integration of Eqs. (2) and (3) are presented in **Table 21**.
>
> 4. Requested Changes (minor) (2): One relevant study
> * The missing reference is added in **Section 2.1**
>
> 5. Requested Changes (minor) (3): The full-form of the abbreviation "NN" is hardly found
> * We missed to elaborate the meaning of NN and overused the abbreviation. We have now provided the full term at the first use of each section.
>
> 6. Requested Changes (minor) (4): The abuse of notations in Eq. (4) should be avoided
> * We have updated Eqs. (1) and (4) following your suggestion.

---

> > ### Comment · Reviewer_1vVW · 2025-01-27
> > **Reliance on pre-trained model**
> >
> > Thank you, the authors, for addressing my comments. However, the concern about using pre-trained models to extract features has been overlooked and not addressed properly. Could the authors provide further discussion and have an ablation study where the method is trained from scratch without using any pre-trained models? I believe that this point is important (as another reviewer also raised the same concern) to understand the effectiveness of the proposed method.

---

> > > ### Author Response · Authors · 2025-01-30
> > > **[Manuscript updated] Training MML from scratch is tricky and computationally expensive.**
> > >
> > > |                          | ImageNet-S |  CUB |
> > > |--------------------------|:----------:|:----:|
> > > | random chance ($1/C$)    | 0.5        | 2.0  |
> > > | MML trained from scratch | 2.4        | 5.8  |
> > > | MML trained with CLIP (from new Table 3)    | **83.0**       | **75.6** |
> > > |
> > >
> > > Thank you for the feedback. We have included the discussion points in the latest manuscript. **The updated text is marked in violet and found in Sec. 3.3 Training and Sec. 4.2 Web-assisted zero-shot classification - Datasets.**
> > >
> > >
> > > We consider pre-trained image-text encoders to be a prerequisite as training MML from scratch causes the following problems.
> > >
> > >
> > > * **[Unreliable feature similarity for $k$NN retrieval and prototype construction]** The similarity between scratch features provides unreliable relevance metrics, leading to the retrieval of unrelated nearest neighbors and the construction of poor-quality prototypes. We have attempted to train MML from scratch without any pre-trained models, but we find it tricky to train the image-text encoder from scratch and achieve low accuracy.
> > >
> > > * **[Lack of a large data pool to train image-text encoder]** We also suspect that the low performance originates from the lack of training data. Note that _training an image-text encoder from scratch requires a huge data pool in general_, e.g, the CLIP image-text encoder is trained with 400M image-caption pairs. On the other hand, MML with the pre-trained CLIP is trained with 1K-9.6K image-text pairs yet generalizes well on unseen classes by memory replacement.
> > >
> > > * **[Expensive memory synchronization with the backbone update]** If the backbone is updated or fine-tuned during training, _all memory element features are required to be synchronized periodically_, which introduces significant computational costs. In contrast, the pre-trained and frozen backbone allows us to pre-extract all memory items once and keep the features unchanged during training.
> > >
> > >
> > >
> > > As we aim to build a lightweight and generalizable model, we choose to use available pre-trained models. The experimental results show that MML generalizes to unseen classes by training with a small amount of training data. Note that MML is not specific to a certain pre-trained encoder. We show the generalizability of MML with two different pre-trained image-text encoders, CLIP and ALIGN (Tables 14 and 15).

---

> > > > ### Comment · Reviewer_1vVW · 2025-02-06
> > > > **Acknowledge the updated manuscript with training from scratch results**
> > > >
> > > > The reported results for training from scratch underscore a significant limitation of kNN, namely its reliance on effective input features.  I concur with the authors' assessment that the dataset's size precludes self-supervised pre-training for feature extraction. This constraint diminishes the study's applicability to certain critical domains where pre-trained models, such as CLIP, are unavailable.
> > > >
> > > > Notwithstanding this limitation, I commend the authors for their transparent presentation of these results.

---

> > > > > ### Author Response · Authors · 2025-02-12
> > > > > **We have added a new paragraph in the discussion section 4.5 "Feasibility on real-world scenarios" and colored it magenta**
> > > > >
> > > > > We appreciate the feedback and agree with the point. We have clearly stated the mentioned weakness in the discussion section for visibility. We have added a new paragraph in the discussion section 4.5 "Feasibility on real-world scenarios" and colored in the magenta color. Please refer to the new version of the manuscript uploaded.

---

### Decision · Action_Editor_LS9E · 2025-02-23

**Recommendation:** Accept as is

**Comment:**

The reviewers were in agreement about recommending acceptance for this paper. The clear writing and comprehensive experiments were recognized as strengths while similarity to previous work and dependence on pretrained features were raised as weaknesses. The authors clearly addressed each of the reviewers' concerns during the rebuttal phase.

In the camera-ready version, the authors are urged to ensure that the appendix is contained inside the main pdf (after references).

**Audience:**

Researchers working in areas surrounding lifelong learning and modular architectures would likely be interested in the results of this paper.

**Claims And Evidence:**

This paper introduces a memory-modular approach to image classification called memory-modular learner (MML). The proposed architecture and relationship to previous work are explained in detail. Experiments show the benefits of MML on zero-shot, few-shot, fine-grained, and class-incremental classification. Additional analyses evaluate the effect of prototype selection, memory type, memory size, and memory robustness. Although MML's reliance on pretrained encoders was raised as a weakness by the reviewers, this limitation has been acknowledged and clearly signposted by the authors in the updated manuscript. Therefore, claims and evidence are sufficiently supported.